# Membrane Internalization Mechanisms and Design Strategies of Arginine-Rich Cell-Penetrating Peptides

**DOI:** 10.3390/ijms23169038

**Published:** 2022-08-12

**Authors:** Minglu Hao, Lei Zhang, Pu Chen

**Affiliations:** 1Advanced Materials Institute, Qilu University of Technology (Shandong Academy of Sciences), Jinan 250014, China; 2Department of Chemical Engineering, Waterloo Institute for Nanotechnology, University of Waterloo, 200 University Avenue West, Waterloo, ON N2L3G1, Canada

**Keywords:** CPPs, arginine-rich peptide, non-covalent interaction, mechanism of internalization, peptide design, biomedical applications

## Abstract

Cell-penetrating peptides (CPPs) have been discovered to deliver chemical drugs, nucleic acids, and macromolecules to permeate cell membranes, creating a novel route for exogenous substances to enter cells. Up until now, various sequence structures and fundamental action mechanisms of CPPs have been established. Among them, arginine-rich peptides with unique cell penetration properties have attracted substantial scientific attention. Due to the positively charged essential amino acids of the arginine-rich peptides, they can interact with negatively charged drug molecules and cell membranes through non-covalent interaction, including electrostatic interactions. Significantly, the sequence design and the penetrating mechanisms are critical. In this brief synopsis, we summarize the transmembrane processes and mechanisms of arginine-rich peptides; and outline the relationship between the function of arginine-rich peptides and the number of arginine residues, arginine optical isomers, primary sequence, secondary and ternary structures, etc. Taking advantage of the penetration ability, biomedical applications of arginine-rich peptides have been refreshed, including drug/RNA delivery systems, biosensors, and blood-brain barrier (BBB) penetration. Understanding the membrane internalization mechanisms and design strategies of CPPs will expand their potential applications in clinical trials.

## 1. Introduction

Cellular uptake of biologically active molecules is a significant obstacle to developing drug design and controlled drug delivery. For instance, it is hard for the monoclonal antibodies to penetrate the cell membrane and enter the cell due to their large molecular weight. Hence, they are limited to being used to identify and target secretory protein targets on the cell surface. However, the application of monoclonal antibodies is severely constrained by the fact that many potential targets for disease therapy are located inside cells. In addition, emerging therapeutic technologies such as gene therapy also need to address the issue of cellular uptake of nucleic acids (DNA, RNA) and other biomolecules. Viral vectors [1,2,3], and methods such as electroporation, microinjection, and liposome encapsulation [4,5,6,7,8,9], have successfully delivered a wide range of therapeutic agents, including proteins, peptides, and oligonucleotides, to target cells. Still, these methods have certain shortcomings, including inefficient drug delivery, cellular toxicity, poor specificity, etc. Against this backdrop, cell-penetrating peptides (CPPs) are coming into view as promising candidates for drug delivery applications [10,11,12,13]. CPPs are a class of small molecules with strong membrane permeability, which can carry peptides, proteins, nucleic acids, and other macromolecules into cells, opening a new pathway for exogenous substances to enter cells. As a typical kind of CPP, the cationic peptide is distinguished by the presence of basic amino acids such as arginine (Arg) and lysine (Lys) [14]. The essential amino acids are positively charged in physiological pH; thus, they can interact with negatively charged drug molecules and cell membranes through non-covalent interaction, including electrostatic interactions [15].

Of all the cell-penetrating peptides, arginine-rich peptides have attracted the most scientific attention as cationic peptides [16,17,18,19]. Examples include the first discovered membrane penetrating peptide, Tat, a typical arginine-rich membrane penetrating peptide. Human immunodeficiency virus type 1 (HIV-1) Tat protein, secreted from infected cells, can deliver several proteins, including ovalbumin, β-galactosidase, and horseradish peroxidase into cells [20,21,22]. The primary domain in Tat protein responsible for crossing the plasma membrane, the protein transduction domain (PTD), is rich in arginine and lysine residues. Jin et al. [23], Know et al. [24], and Nagahara et al. [25] verified that the protein could serve as a carrier to direct the uptake of heterologous proteins into cells by generating genetic in-frame PTD fusion proteins. Moreover, Park et al., showed that the smallest structural domain of the Tat protein that acted as a membrane penetrator was the amino acid sequence at positions 49–57 (residues 49–57: RKKRRQRRR), which was a 9-amino acid sequence containing 6 arginines [26].

Numerous investigations have been conducted to investigate how CPPs transport substances into cells. The two primary recognized cellular uptake mechanisms are endocytosis and the pore formation model. [13,27,28,29,30,31,32,33,34,35] (Figure 1A). The ratio between endocytosis and direct cell entry is critical in the cellular uptake of cell-penetrating peptides, and the modification of the CPPs will affect the ratio between endocytosis and direct translocation. While the mechanisms of how the modification affects the ratio and the effect of the following details have not been fully understood, Zhang et al. [36] speculated that the positively charged arginine on the periphery of the NP1 peptides could greatly facilitate their direct translocation through the negatively charged plasma membrane via electrostatic interaction instead of via endocytosis, which provides a more efficient uptake pathway. On the other hand, the arginine and lysine residues also significantly impact the transmembrane function of cationic CPPs [14]. However, there are several hypotheses about the involvement of arginine and lysine in membrane penetration. In this review, we highlight the role of arginine residues in the arginine-rich CPPs and the effects of arginine-rich peptide structural alterations on their functions. Moreover, we provide an update on the progress of arginine-rich peptides in biological applications. By elucidating the function of arginine in CPPs or the structure of transmembrane efficiency, this arginine-rich delivery platform could be made even more specialized by applying logical design to meet the desired biological applications.

**Figure 1 ijms-23-09038-f001:**
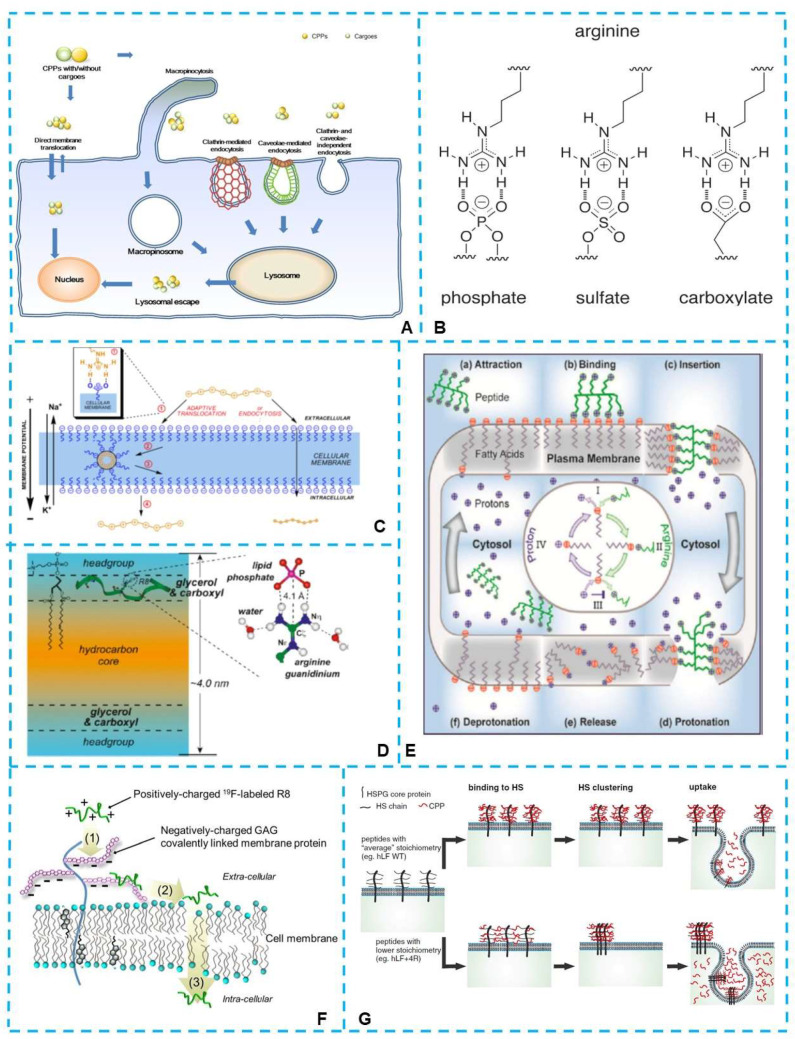
(**A**) Different cellular entry routes for either cell-penetrating peptides (CPPs) alone or CPP/cargo complexes. Direct membrane translocation and endocytosis were two major routes which had been proposed [33]. (**B**) Possible divalent hydrogen bond formation by the side-chain guanidino moiety of arginine with phosphate, sulfate, and carboxylate groups [37]. (**C**) Mechanisms of the arginine-rich CPPs uptake. The guanidinium group formed a bidentate bond with negative phosphates, sulfates, and carboxylates on the cell surface [38]. (**D**) Mechanism of the phosphates-related membrane penetration of arginine-rich CPPs, HIV Tat (48–60) [39]. (**E**) The proposed cellular uptake mechanism for arginine-rich CPPs revealed that deprotonated fatty acids were involved in the membrane penetration process [40]. (**F**) The mechanism for non-endocytic, energy-independent translocation of 19F-R8 into cells. The mechanism involved (1) binding of 19F-R8 to GAG at the cell surface, followed by (2) the transfer to the cell membrane and (3) the entry into the cytosol [41]. (**G**) Schematic of the proposed stoichiometry-dependence of uptake. The guanidine groups in arginine formed bidentate hydrogen bonds with negatively charged heparan sulfates. The cross-linking of HS was the driving force for the uptake of arginine-rich CPPs [42].

## 2. Cell-Surface Interactions on Arginine-Rich CPPs Allow for Internalization

### 2.1. Binding to Anionic Groups to Promote Uptake

#### 2.1.1. Binding to Phosphate Anionic Groups

Numerous studies have highlighted that guanidinium-rich arginine is essential for cellular uptake. Since arginine-rich peptides preferentially bind to negatively charged molecules, the cell membrane penetration is assumed to initiate by binding to various anionic groups via non-covalent interaction, such as phosphates, carboxylates, and sulfates (shown in Figure 1B), on cell surfaces [37]. In the early work of Wender et al., the function of multiple guanidine groups in arginine-rich CPPs in membrane penetration was summarized: the guanidine group of arginine formed a bidentate bond with negative phosphates, sulfates, and carboxylates on the cell surface, and the charge-neutralized species were driven into the cell by the membrane potential [17,38] (Figure 1C). The transmembrane mechanism of arginine-rich CPPs was further investigated in the following decades.

For an exploration of the phosphates-related membrane penetration mechanism of arginine-rich CPPs, HIV Tat (48–60), a solid-state NMR technique, was applied. The results suggested that Tat was inserted into the glycerol backbone region of the membrane-water interface in anionic lipid bilayers and generated transient membrane defects, which relied on transient interactions between the arginine side chains and lipid phosphates [39] (as shown in Figure 1D). In another study, Chen et al. [43] found that steady-state water holes occurred when the guanidinium groups in the arginine-rich Tat associated with the phosphate moieties on the lipid headgroups, and Arg residues pulled down water molecules in the membrane to stabilize the insertion. Meanwhile, bidentate hydrogen bonding involving Arg residues with the lipid phosphate groups was observed in the study of Jobin et al. [44]; the non-covalent interaction could induce invagination phenomena and contribute to tubulation and internal vesicle formation.

#### 2.1.2. Binding to Carboxylate Anionic Groups

Binding to carboxylates is also a common way of promoting the membrane penetration of the arginine-rich CPPs. In the study of Herce et al., an efficient energy-independent translocation mechanism for arginine-rich molecules was revealed [40]. A transient membrane channel was formed when the cell exterior guanidine groups were attracted and bound to the deprotonated fatty acids on the membrane. In the crossing process of the peptide−fatty acid complex, the protons from the cytosolic side competed for the binding of the guanidinium groups to fatty acids, and the high density of protons in the cytosol protonated the fatty acids, which resulted in the release of the peptide into the cytosol (shown in Figure 1E).

#### 2.1.3. Binding to Sulfate Anionic Groups

As described above, sulfate is also one kind of critical anionic group binding to the arginine-rich peptides. Sulfated glycosaminoglycans (GAGs), such as heparan sulfate and chondroitin sulfate localized on cell membranes, are covalently linked to core proteins at the cell surface [18,27,41,45,46,47]. Real-time in-cell NMR spectroscopy was applied to investigate the direct membrane translocation of 19F-labeled octaarginine (R_8_) into living cells in the work of Takechi-Haraya et al., and they found that arginine-rich 19F-R_8_ entered hydrophobic cell membrane after binding to GAGs [41] (Figure 1F). In the authors’ opinion, the process of the membrane penetration of 19F-R_8_ was non-endocytic and energy-independent. One possibility was that the charge neutralization of polyarginine with GAGs induced insoluble and energetically unstable peptide-GAG complexes, which led to the dissociation of 19F-R_8_ from GAGs to water or rapidly transferred to the cell membrane. In another study, Takechi-Haraya et al., investigated several arginine-rich peptides of Tat, R_8_, and Rev, with heparin as a GAG model; they found that the beneficial electrostatic interaction between arginine residues of peptide and anionic sulfate/carboxyl groups of heparin contributed to the favorable enthalpy gain, which served as an energy source to facilitate their cell penetration [48]. On the other hand, several studies have proven that GAG cross-linking has been associated with the activation of signaling pathways that result in endocytosis [49,50,51]. Zuconelli et al., investigated the role of calcium in direct cytosolic uptake of two arginine-rich peptides, nona-L- and D-arginine. They indicated that the calcium channel Orai1 played a decisive role in triggering rapid uptake of the peptides [52].

Binding to negatively charged heparan sulfates (HS) at the cell surface is considered the first step in the internalization of CPPs. Wallbrecher et al., investigated a collection of the lactoferrin-derived CPPs with respect to HS binding and uptake; the study demonstrated that the guanidine groups in arginine could form bidentate hydrogen bonds with negatively charged heparan sulfates, and the cross-linking of HS was the driving force for the uptake of arginine-rich CPPs [42] (Figure 1G). Interestingly, peptides with a low stoichiometry had a higher capacity to cross-link HS, which did not hold for other classes of CPPs [53]. Ikuhiko [54] found that the arginine-rich peptides led to the activation of small G-protein Rac1 and reorganization of actin (lamellipodia and membrane ruffling), which improved the cellular uptake of the peptides and their cytosolic translocation. In another study by Kawaguchi et al., syndecan-4, one of the heparan sulfate proteoglycans, was demonstrated to be a primary cell-surface receptor involved in the endocytosis of the octa-arginine (R_8_) peptide [51].

### 2.2. Participate in Membrane Penetration by Partitioning the Lipid Glycerol Regions

Besides binding to the anionic groups on cell surfaces, the guanidine group participates in membrane penetration by partitioning the lipid glycerol regions. According to widespread consensus, arginine-rich CPPs are extremely cationic and hydrophilic; their guanidine groups create ion and hydrogen interactions with lipid head groups to assist membrane binding. During the membrane translocation of the CPPs, the peptide backbone must pass through the lipid core region. Hydrophobic interactions between the lipid core and fewer hydrophilic peptide backbones could lead to translocation due to poorly hydrophilic methylene groups in the side chains of arginine and the peptide backbone. Umbrella sampling simulations were used in the study of Sun et al. [55]; the arginine-rich peptide octa-arginine (R_8_) was found to expand the surface area of the lipid bilayer due to the deep partitioning of guanidinium ions into the lipid glycerol regions. Meanwhile, R_8_ was also found to extend the lifetime of the transient membrane pore due to inserting an arginine side chain into the existing pore. The study demonstrated that the arginine-rich peptide was essential in membrane pore formation and lifetimes. In addition, the study of Pourmousa et al. [56] demonstrated that H-bonds and charge-pair interactions between the bilayer and arginines and lysines were the critical interactions in the binding mode of penetrating and helped charged residues to localize in the hydrophobic region of the bilayer.

## 3. Peptide Design of the Arginine-Rich CPPs

### 3.1. Primary Sequence Design of the Arginine-Rich CPPs

#### 3.1.1. Number of Arginine Residues in the Peptide Sequences

As mentioned above, arginine plays an essential role in the functional performance of delivery peptides, and the number of arginine residues in the peptide sequences affects the intracellular delivery efficiency of the arginine-rich CPPs. Previous studies revealed that peptides with 7−15 arginine residues had the most efficient cellular uptake [57,58,59], and the following works of Nakase et al. [60] and Kosuge et al. [58] proved that point. In their study, oligoarginine peptides with different numbers of arginine residues (Rn: *n* = 4, 8, 12, 16) were used to modify the extracellular vesicles (Figure 2A). The results indicated that the cellular uptake and cytosolic release efficiency were strongly affected by the number of arginine residues and that the conjugation of R_8_-EVs or R_12_-EVs resulted in the highest cellular uptake of all the oligoarginine-peptide conjugations. The efficiency decreased when the number of arginines came to 16, which means that the number of arginines was not the more, the better. Dafeng et al. [61] and Dong-Wook et al. [62] demonstrated that longer oligoarginines had a more remarkable ability to aggregate siRNA than shorter oligoarginines. In another study, Naggar et al., revealed that dfR6, dfR7, and dfR8 displayed more robust cytosolic penetration and nucleolar staining compared with dfR4 and dfR5, which once again illustrated that the number of arginine residues was important to the performance of CPPs [63].

#### 3.1.2. Arginine Optical Isomers in the Peptide Sequence

Isomer is another factor affecting the function of CPPs [10,64,65,66,67,68]; a different number of arginine optical isomers in the peptide sequence are proved to affect the uptake efficiency of the delivery peptides. Verdurmen et al. [69] firstly reported the differences in the uptake of L-CPPs (R_9_) and their D-counterparts (r_9_); the results demonstrated that cationic L-CPPs (R_9_) were taken up more efficiently than their D-counterparts (r_9_) in MC57 fibrosarcoma and HeLa cells but not in Jurkat T leukemia cells (the sequences of the peptides were shown in Table 1). For further and comprehensive research of the D- and L-arginine residues in CPPs, Ma et al. [70] developed a series of novel chimeras consisting of various numbers of D- and L-arginine residues. The results demonstrated that the number of D-Arg residues in the CPPs significantly affected their cellular uptake behaviors (Hela, HEK293, RAW264.7, and EL4 cells were used in the study). When there were fewer than 3 D-Arg residues in the peptide backbone, increasing the number of D-Arg residues resulted in a diffuse cytosolic peptide distribution at low concentrations and higher uptake efficiency [R_8_, rR_7_, (rR)_2_R_4_, and (rR)_3_R_2_]. The further replacement of L-Arg residues with D-Arg residues did not significantly affect the intracellular distributions when there were more than 3 D-Arg residues [(rR)_3_R_2_, (rR)_4_, r_2_(rR)_3_, and r_8_]. Moreover, the uptake efficiencies of the octa-arginines containing 3 or more D-Arg residues [(rR)_3_R_2_, (rR)_4_ and r_2_(rR)_3_] were comparable to that of r_8_ and much better than that of R_8_. Interestingly, Behzadi et al. [71] explored the selectivity and cellular uptake of HHC36, a Trp/Arg-rich nonapeptide, and its D-enantiomer ((all D)HHC36). The results unraveled that the D-handedness-selective toxicity of the Trp/Arg-rich sequence was associated with the cell type.

#### 3.1.3. Position of the Arginine Residues in the Primary Sequence

The amino acid sequence on the primary structure and the aggregated three-dimensional conformation on the secondary structure both significantly impact the arginine function. As aforementioned, the number of arginine residues in the peptide sequences affect the intracellular delivery efficiency of the arginine-rich peptide. When the amount of arginine remains the same, but only the position of the arginine amino acids in the sequence is changed, will the function of the arginine-rich CPP change? The work of Konate et al. [79] revealed the answers for us. To maintain the same charge number as the parental peptides, they created peptides with two more arginine residues at both ends but two fewer arginine residues in the middle of the sequence (Arg  =  4 for W1, W5, W1-4R, and W5-4R). The peptides showed significant differences in stability, the ability to form peptide-based nanoparticles and luciferase silencing activity. The peptides with the 5-residue consensus motif, LRLLR at positions 5–9, were created and compared in work by Fuselier et al. [80]. The results suggested that multiple primary sequences of leucine and arginine could support spontaneous membrane translocation, and sequence context was necessary for the function of the peptides.

### 3.2. Secondary Structure of the Arginine-Rich CPPs

A stable helical structure of the arginine-rich peptide is necessary for their cell-penetrating abilities [81,82], causing the formation of the α-helical structure to decrease the polarity and the free energy cost of transfer to the hydrophobic membranes for peptide bonds in the peptide molecule [83]. To gain insight into the significance of α-helix formation for transcellular cargo delivery, Komin et al. [72] designed a peptide (RLLrLLR, CLr) with the same sequence as the CL peptide (RLLRLLR). Still, it could not form an alpha helix, and the results showed that the peptide with α-helix formation had a higher transcellular transport capacity even though they had the same sequence. To study the effect of arginine residues on the cell-penetrating ability caused by secondary structure changing, Ohgita et al. [73] designed a novel CPP, A2-17 (LRKLRKRLLRLWKLRKR). A2-17 had a lower amount of arginine but a higher cell penetration effectiveness than a typical arginine-rich CPP, Rev (TRQARRNRRRRWRERQR), which could be attributed to the creation of an amphipathic α-helix and an increase in amphipathicity of A2. The circumferential distribution of arginine residues in Rev. interfered with the development of the -helix after binding to lipid membranes. In amphipathic A2-17, however, such charge repulsion from arginine residues could be offset by the free energy reduction following lipid binding [84], facilitating the transfer of peptides from GAGs to lipid membranes and boosting membrane penetration efficiency. The work of Yamashita et al., also confirmed the importance of the α-helical structure to the efficiency of cell penetration using a cationic cyclic α, α-disubstituted a-amino acids: Api^C2Gu^ (which possessed an arginine mimic side chain). They found that the peptide containing Api^C2Gu^ formed a stable α-helical structure and was more effective at penetrating cells than the nonhelical Arg nonapeptide (R_9_) [85]. Moreover, Chen et al. [86] found that contributing to the helical structure formed by the positive charge of arginine residues in one face of the structure and the anionic component in siRNA, the peptide-siRNA complex had a better interaction with cell membranes, and the cellular internalization was enhanced. As mentioned above, GAGs played a crucial role in the cellular uptake of cationic CPPs; Neree et al. [87] found that an α-helix was a crucial step for GAGs-mediated endocytosis of pituitary adenylate cyclase-activating polypeptide (PACAP, its sequence and structure were shown in Figure 2B), which indicated that the uptake efficiency of a given cationic CPP was not only correlated to its ability to bind to or cluster GAGs but also affected by the peptide conformation.

### 3.3. Ternary Structure of the Arginine-Rich Peptides

It was previously demonstrated that the cytotoxicity of high concentrations of CPPs, as well as the breakdown of CPPs by proteases during cargo transport, hampered the development of effective CPPs. While reducing the length of CPPs appeared to be a viable way to lessen peptide toxicity, restricting the number of charges (below eight) reduced the transmembrane efficiency of CPPs [88]. Based on this background, a method combining very short CPP sequences and collagen-like folding domains was designed [89]. The CPP domains (RRRRRR or RRGRRG) and multiple proline-hydroxyproline-glycine (POG [proline-hydroxyproline-glycine])*_n_* were combined to form a collagen-like triple-helical conformation (shown in Figure 2C). The folded peptides with CPP domains had low cytotoxicity and higher stability to enzymatic degradation while retaining high membrane penetration efficiency. From the above study, we could see that it was conducive to increasing the peptide chain length and improving the resistance of the peptides to enzymatic degradation by constructing CPPs with folding domains. The study of Oba et al., further supported this conclusion [90]. They prepared alpha-Aminoisobutyric acid (Aib) CPP foldamers (Arg-Arg-Aib)(n) (*n* = 1–6) for pDNA transfection, and the results showed that longer peptides (*n* ≥ 4) exhibited better transfection abilities than an Arg nonapeptide. Furthermore, Aib CPP foldamers demonstrated better transfection abilities attributable to the high resistance of the peptides to enzymatic degradation. Besides folding domains, other designs of the CPPs were proved to have excellent delivery efficiency. Yoo et al., designed a branched poly-CPP structure (Figure 2D) of nona-arginine (mR9) and synthesized a branched-mR9 (B-mR9) using disulfide bonds to deliver nucleic acid molecules [91]. The branched structures improved the transfection capability owing to the strong electrostatic attraction between DNA and siRNA molecules. In addition, the branched-mR9 exhibited less cytotoxicity compared to conventional CPPs.

### 3.4. Modification of the Arginine-Rich CPPs

#### 3.4.1. Improve the Transmembrane Capacities of Arginine-Rich CPPs via Hydrophobic Elements

Commonly, the cationic arginine-rich CPPs are non-amphipathic; when they are covalently or non-covalently associated with hydrophobic hydrocarbon moieties, the physicochemical properties of the cationic CPPs will be dramatically changed by these hydrophobic entities. The addition of a hydrophobic counterpart makes the CPPs become primary amphipathic molecules (their primary structures contain well-defined cationic and hydrophobic domains), which increases their affinity for neutral or weakly anionic membrane bilayers because they can interact both with the phosphate groups of the phospholipids (PLs) and with the hydrophobic core of the membrane bilayer [92,93,94], and then penetrate the bilayer inducing phase separation, permeabilization, and disruption of the membrane [95] (Figure 2E). Interestingly, the addition of a single hydrophobic residue to an Arg-rich peptide could drastically modify the translocation mechanism [96]. It might be relevant to the participation of these hydrophobic counterparts in the formation of inverted micelles. In other studies [97,98], amphiphilic counteranions (pyrenebutyrate counteranion, PyB) were found to have the ability to promote the internalization of arginine-rich CPPs. The presence of PyB induced the disruption of liquid-ordered domains, which enhanced the overall fluidity of the membrane [99] and improved the translocation ability of oligoarginines. Noteworthy, the hydrophobic component could be introduced by using unnatural amino acids; the arginine-rich CPPs with the structure of (Arg-X-Arg)*_n_* were designed. The insertion of 6-aminohexanoic acid (Ahx) improved the delivery efficiency of nucleic acids and the metabolic stability of a designed CPP, stearylated (Arg-Ahx-Arg)_4_ [100]. Futaki et al. [101] found that the attachment of a series of penetration accelerating sequences (Pas), especially phenylalanine (Phe) residues, could enhance the cellular uptake of arginine-rich CPPs, R_8_. As we mentioned above, the addition of Phe residues increased the hydrophobicity of the CPPs and thus its affinity for membranes. Moreover, the addition of Phe residues might affect the actin rearrangement and membrane ruffling, inducing a higher ability of micropinocytosis. Meanwhile, the attachment of Phe residues might further increase peptide−proteoglycan interactions, resulting in increased stimulation of micropinocytosis and the partition of peptides into membranes.

Acylation was proved an efficient way to improve the cell-penetrating properties of the arginine-rich peptides. The study of Oh et al. [102] demonstrated that a combination of acylation by long chain fatty acids and cyclization on short arginine-containing peptides could improve their cell-penetrating property, possibly through efficient interaction of rigid positively charged R and hydrophobic dodecanoyl moiety with the corresponding residues in the cell membrane phospholipids. On the other hand, the study of Swiecicki et al., also proved that the unsaturated acyl chain promoted the short oligo-arginine lipopeptide membrane translocation and endocytosis [103].

In recent research, 4-(dimethylaminoazo) benzene-4-carboxylic acid (DABCYL) was used as a new kind of modification because of its structural simplicity, availability along with hydrophobicity, which could assist cell permeability. It has been demonstrated that utilizing the DABCYL group to modify arginine-rich CPPs effectively increases cellular uptake efficacy [104,105,106].

**Figure 2 ijms-23-09038-f002:**
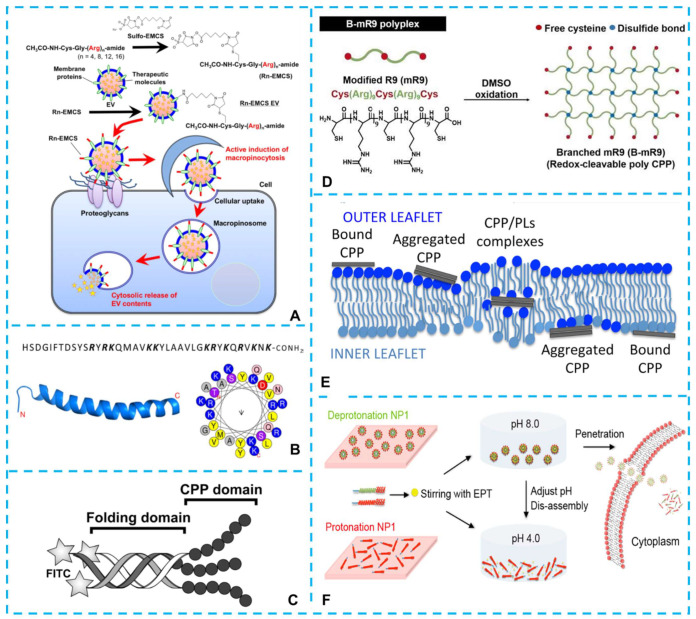
(**A**) Oligoarginine peptides with different numbers of arginine residues (Rn: n = 4, 8, 12, 16) were used to modify the extracellular vesicles [60]. (**B**) Sequence and schematic ribbon representation of PACAP secondary structure and helical wheel representation of the putative a-helix segment of PACAP [87]. (**C**) Schematic representation of the CPP folded into a triple helix within the (POG)n sequence (folding domain). The CPP domain R6 or (RRG)_2_ was located at the C-terminus [89]. (**D**) Schematic illustration of the synthesis of the branched-modified R9 (B-mR9) CPP [91]. (**E**) Model of CPP direct translocation via the formation of inverted micelles. In this model, CPPs were internalized as neutral and hydrophobic complexes with anionic phospholipids (PLs) [CPPp+(PL-)p] [95]. (**F**) Schematic diagram of the pH-triggered self-assembly/disassembly of peptide NP1 to encapsulate and deliver the cancer drug ellipticine into cancer cells [36].

#### 3.4.2. Modification of the Arginine-Rich Peptides via Other Amino Acids

In the arginine-rich CPPs, arginine residues play a critical role in the structure and function of the peptides. In addition, other amino acid residues can also affect the delivery efficiency of the CPPs. In the study of Zhang et al. [36,74], histidine with an imidazole side chain served as a hydrogen donor or acceptor. This property of pH-dependent hydrogen bonding (or positively charged) was used to design a self-assembly/disassembly pH-triggered peptide (NP1, stearyl-HHHHHHHHHHHHHHHH-RRRRRRRR-NH_2_). The NP1 (at pH 8.0) could encapsulate and deliver the cancer drug ellipticine to A549 and CHO-K1 cells (Figure 2F). Walrant et al., demonstrated that Trp residues played the role of natural aromatic activators of Arg-rich CPPs. The presence of Trp in oligoarginines increased the uptake in cells expressing GAGs at their surface [107]. In another study, nona-arginine (R_9_) and a small peptide containing 6 arginine and 3 tryptophan residues (RW9) were used to explore the penetration of the two peptides into large unilamellar vesicles. Compared with R_9_, the amphipathic peptide RW9 crossed the membrane vesicles, thus demonstrating an important role in the increasing membrane fluidity and peptide membrane translocation of the Trp residues induced amphipathy [108]. To explore the influence of the tryptophan content and backbone spacing on the CPP uptake efficiency, peptides with different tryptophan contents and compositions were explored in the study of Rydberg et al. [109]. They found that the peptides with additional tryptophan content and backbone spacing had further intracellular distribution. The peptides with four tryptophans in the middle or evenly distributed along the peptide sequence had higher uptake efficiency than those with four tryptophans at the N-terminus, even though the peptides had the same amino acid content. In the study of Fuselier et al. [80] we mentioned above, the leucine residues played an important role in the spontaneous membrane translocation of the peptides containing the LRLLR sequence.

## 4. Applications of Arginine-Rich Peptides in Biomedicine

### 4.1. Application of Arginine-Rich Peptides in Drug Delivery

#### 4.1.1. siRNA Delivery

In the past decades since the Nobel prize-winning discovery of RNA interference (RNAi), RNA interference has become a valuable research tool for studying gene function, regulation, and therapy. Since the poor stability and inability of naked siRNA to translocate through cell membranes, an appropriate siRNA delivery material is highly desired to transport short interfering RNA (siRNA) to the target sites to achieve the clinical potential of RNAi for specific gene silencing. Many methods for delivering siRNA have been explored in the past few years. However, the lack of a cell type-specific, safe, and efficient delivery material is still the primary bottleneck for the clinical application of siRNA. Among the emerging candidate nanocarriers for siRNA delivery, peptides have drawn attention due to their structural and functional versatility, potential biocompatibility, and ability to target cells. The cationic arginine-rich peptides could deliver siRNAs into targeting cells and induce silencing of the therapeutic target contributing to its membrane penetration capacity. The 599 peptides (GLFEAIEGFIENGWEGMIDGWYGGGGRRRRRRRRRK) designed by Cantini et al. [75] proved that the cationic CPP residues could enhance the intracellular delivery and bioavailability of siRNAs. On the other hand, the hybrid nanovector (based on PEGylated superparamagnetic iron oxide nanoparticles functionalized with gH625 peptide, chitosan, and poly-L-arginine) designed by Sanaa et al. [110] demonstrated that the cationic polymers (poly-L-arginine) could provide siRNA protection and favor siRNA endosomal escape and delivery to the cytosol.

#### 4.1.2. Anti-Cancer Drugs Delivery

Typically, chemical cancer drugs transport cell membranes via a passive diffusion route, which is significantly affected by drug concentrations and chemical-physical properties of cell membranes [111,112]. Thus, the peptide-based drug delivery systems provide a promising perspective for effectively delivering chemical drugs to fight cancer. Arginine-rich CPPs and ligands specific to target over-expressed receptors on cancer-cell surfaces are popular among many approaches for active cancer therapy. Deshpande et al. [113] proved that the attachment of an arginine-rich CPP octaarginine (R_8_) and transferrin (Tf) to the surface of DOX-loaded liposomes improved the targeting of A2780 ovarian carcinoma cells and controlling tumor growth in an A2780 ovarian xenograft model. Besides ovarian carcinoma [114], the arginine-rich CPPs also showed their application potential in the therapy of other cancers, such as colorectal cancer [115,116,117], lung cancer [36,118,119,120,121,122,123], oral cancer [75,124,125], breast cancer [122,126,127,128,129], prostate cancer [130,131,132,133], pancreatic cancer [134,135], renal carcinoma [121,136], etc.

Besides the application in the peptide-based drug-delivery systems, the arginine-rich peptides are also used in other related fields of cancer therapy. For example, anti-angiogenic therapy is a potential chemotherapeutic strategy for treating drug-resistant cancers but lacks suitable methods for delivering such drugs to tumor endothelial cells. Takara et al. [136] designed liposomes modified by NGR (CYGGRGNG) motif peptide and tetra-arginine (R4) peptide, which enhanced the amount of delivered liposomes, induced the disruption of tumor vessels and suppressed tumor growth.

Regulatory T cells (T_reg_) have been proved to hinder protective immunosurveillance of neoplasia and hamper effective antitumor immune responses in tumor-bearing hosts, thus promoting tumor development and progression [137,138,139,140,141,142]. The arginine-rich peptides used in the T_reg_ related tumor immunity therapy were explored by Morse et al. [143]. The arginine-rich CPP (RXR)_4_ was conjugated with morpholino oligomer. The conjugate could modulate T_reg_ levels and enhance the induction of effector T-cell responses, potentially optimizing immunotherapy strategies in cancer and viral immunotherapy.

#### 4.1.3. Targeted Delivery Properties

The arginine-rich CPPs are found with excellent prospects in the field of drug delivery; however, the nontargeting features of CPPs greatly limited their application in systemic administration. Here, one solution for building the targeted delivery of CPPs is modifications of the arginine-rich peptides via targeting compounds, where the strategy is similar to other targeting delivery systems [144]. Hence, conjugating specific motifs to the arginine-rich peptides to form tandem peptides has been proved an effective method [33,145]. Chen et al. [146] designed two self-assembled BolA-like arginine-rich peptides as viral-mimetic gene vectors. In the structure of the peptide, the R_8_ sequence was designed as the cell-penetrating moiety and the arginine-glycine-aspartic acid (RGD) sequence as tumor-targeting moiety (Figure 3A). In the study of Liu et al. [147], a formed tandem peptide R6-dGR had been designed and proved to have tumor targeting and penetrating properties both in vitro and in vivo. Besides conjugating specific motifs, the arginine-rich peptides could also be modified by other functional groups to form multifunctional nanomaterials, which opened new frontiers for the applications of arginine-rich peptides in targeted delivery. For example, Ji et al. [148] developed a dual-mode nanomaterial with the ability of cancer-associated fibroblasts (CAFs) to target and efficient cell penetration to achieve tumor-targeted drug delivery (Figure 3B). The core-shell structured peptide nanoparticles (PNP) had a hydrophobic cholesterol core and a hydrophilic cationic R_9_ peptide shell. Afterwards, hydrophobic antitumor drug doxorubicin (DOX) was encapsulated by PNP, and then mouse monoclonal antibody (mAb) molecules targeted human fibroblast activation protein-α (FAP-α) were modified onto the surface to construct PNP-D-mAb nanoparticle. In another study, a nanoparticle with a similar self-assembled core-shell structure was designed with a multi-functional fusion peptide sequence (GFLGR_8_GDS) and a hydrophobic polycaprolactone (PCL) tail [149]. The peptide sequence was located on the shell and encapsulated the DOX in the core. The RGD and membrane-penetrating peptide (R_8_) sequences in the nanoparticles could be used to target tumor cells and penetrate cell membranes (Figure 3C). As seen from the above studies, the RGD sequence was an influential tumor-targeting segment. An amphiphilic peptide (RR-22) with a sequence of Ac-RGDGPLGLAGI_3_GR_8_-NH_2_ was designed for the aim of selective cancer-killing (Figure 3D) [76]. Its high cancer-killing selectivity was ascribed to the specific recognition and binding of RGD segment to cancer membranes and the cleavage of PLGLA segment by the cancer-overexpressed matrix metalloproteinase-7 (MMP7).

Besides the popular RGD peptide, other targeting identifications of tumor could be recognized as targeting sites and development of the CPPs [150,151,152]. New peptide sequences with targeting properties are emerging to be discovered and applied for practical purposes, which shall facilitate the development of next-generation therapeutic molecules with both selectivity and targeting ability.

### 4.2. Application of Arginine-Rich Peptides in Biosensors

The arginine-rich CPPs have the potential for widespread use in the field of biosensors due to their exceptional cellular uptake capacities. Researchers have used arginine-rich peptide Tat as a delivery tool for biosensors for over 10 years. A ratiometric fluorescent zinc biosensor was created [153] to better comprehend zinc levels in conventional cells. Tat was fused to human carbonic anhydrase and employed as a sensor transducer. This allowed the construct to be internalized effectively without the need for cell membrane manipulation. In order to map the pH of the whole cell and assess changes in the pH of the cytoplasm and lysosomes simultaneously, Xia et al., developed a novel ratiometric fluorescent probe based on arginine-rich peptides [154]. The arginine-rich CPP, R_12_K, worked as a linker, carrier, and part of the fluorophore. In another work of their team, R_12_ was used to modify the optical properties of 5-carboxylfluorescein (FAM), a conventional pH-responsive fluorophore widely used in fluorescence labeling and imaging, presenting a potential method for producing fluorescent pH probes with adjustable pK_a_ values for measuring the pH in organelles [155]. The water solubility, membrane permeability, and organelle-specific localization of FAM were all enhanced by the arginine-rich CPPs in addition to tunable pK_a_. In some earlier studies, arginine-rich peptides were used in the application of optical calcium sensors [156] and high-throughput screening of inhibitors of protein kinases [157]. Moreover, the arginine-rich peptides were also used in the field of oncotherapy-related biosensors. In prostate cancer cell lines, an arginine-rich cell-permeable peptide (NH_2_GR_11_) was discovered with an unanticipated preferential uptake. The peptide was employed as a positron emission tomography (PET) imaging probe for the targeted detection of distant prostate cancer metastases [158].

Applications of arginine-rich peptides in biosensors have received more and more attention due to their high effectiveness at membrane penetration. Focusing on the applications of biosensors from different fields could assist researchers in broadening their perspectives and laying the groundwork for efforts to identify new and better therapeutic options.

### 4.3. Application of Arginine-Rich Peptides in Antimicrobial

Due to the quick and broad-spectrum antibacterial activity and apparent decreased risk of microbial resistance development, antimicrobial peptides (AMPs) have drawn as much attention as novel families of antibiotics. Interestingly, recent findings suggest that CPP conjugation to AMPs may be useful for increasing antimicrobial activity and selectivity against bacteria. The study of Lee et al. [159] was the first study to investigate the effects of CPP−AMP conjugates on antimicrobial activity, as well as their mode of action. They found that CPP (R_9_) conjugation to AMPs facilitated translocation across the membrane and entry into bacterial cells, and the conjugates showed stronger anti-inflammatory activity than the AMPs alone (Figure 3E). In the study of Kravchenko et al. [78], hybrid peptides R23F, R23DI, and R23EI (sequences in Table 1) based on the ribosomal S1 protein sequence from S. aureus and with an arginine-rich CPP fragment (RKKRRQRRR) in the N-terminus of the peptides were synthesized and showed antibacterial activity. In addition, Grishin et al. [160] and Kurpe [161] also designed and synthesized hybrid peptides containing arginine-rich CPP fragments in the N-terminus of the peptides, which opened up new possibilities for the manifestation of the antimicrobial effects of hybrid peptides.

### 4.4. Application of Arginine-Rich Peptides in Blood-Brain-Barrier Transport

The intrinsic neuroprotective mechanism of the human brain makes it difficult to transport biotherapeutics across the blood-brain barrier (BBB). Recent research suggests tailored nanoparticles might be created using cationic cell-penetrating peptides for targeted biotherapeutics’ delivery to the brain. The capacity of arginine-rich CPPs through the blood-brain barrier was simulated and verified by researchers. In the study of Rousselle et al., the potential of the anti-cancer drug doxorubicin (Dox) to cross the BBB was investigated using a rat brain model after it was linked to the peptides SynB1 (RGGRLSYSRRRFSTSTGR) [77]. The outcomes showed that Dox uptake increased by a factor of 6 when combined with SynB1 vectors, suggesting that SynB1 enhances Dox delivery across the BBB. Sharma et al. [162] reported a bifunctional liposome combining transferrin (Tf)-mediated receptor targeting and poly-L-arginine-facilitated cell penetration, which showed high efficiency and low toxicity in the in vitro blood-brain barrier model (Figure 3F). In another study, 11 poly-arginine (11R) was proved capable of in vivo delivery to the brain by passing through the BBB, allowing for the administration of therapeutic compounds in cerebral ischemia [163]. Moreover, Tian et al. [164] designed a multifunctional polymeric micelle showing satisfactory properties of BBB penetration and tumor targeting in both xenograft and orthotropic glioma mouse models, in which arginine-glycine repeats (RG)_5_ were incorporated into the micelles to improve cellular uptake and across the BBB. In a word, the capacity of arginine-rich CPPs through the BBB could be a promising strategy for safe and effective drug delivery to the brain.

Due to the unique side chain of arginine with a positive charge, the CPPs provide not only cell penetration ability but also adjust the charge of the whole peptide sequences, which significantly expands its application in various fields besides biomedical applications, such as food science [165,166], materials science, engineering [167], etc. Along with the development of the design strategies of CPPs and the deep understanding of the peptide-membrane interactions, it will bring enormous opportunities for the wide applications of CPPs in the future.

**Figure 3 ijms-23-09038-f003:**
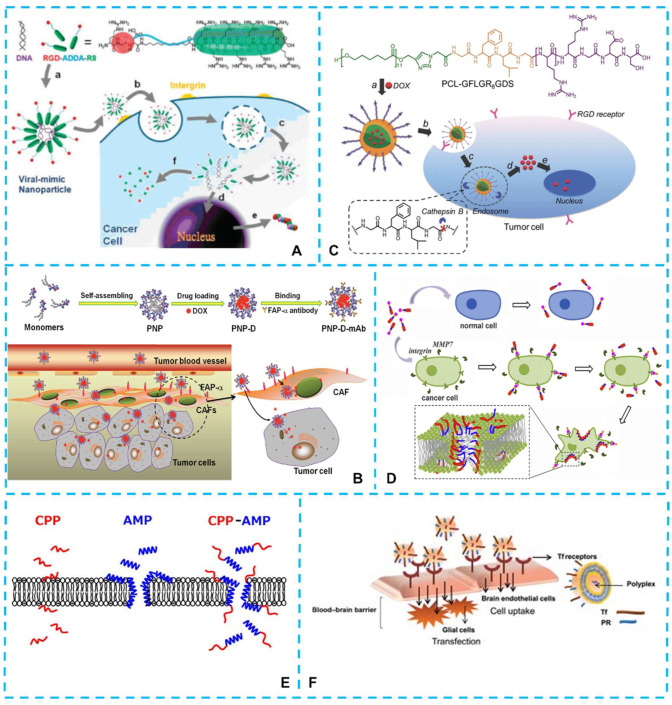
(**A**) BolA-like amphiphilic peptides attach to plasmid DNA to self-assemble viral-mimetic nanoparticles with low cytotoxicity and high gene transfection efficiency [146]. (**B**) Schematic illustration of the nanoparticle formation process, including peptide assembling, drug loading, and mAb modification, and the proposed mechanism of PNP-D-mAb in CAFs targeting and drug penetration [148]. (**C**) Schematic illustration of the self-assembly of the peptide (PCL-GFLGR_8_GDS) to load the anti-tumor drug DOX for targeted cancer therapy [149]. (**D**) Schematic representation of the selective cancer-killing processes of RR-22. The specific RGD segment can recognize and bind to cancer membranes, and the overexpressed MMP7 (not available in normal cells) can cleave RR-22 to release the active cell-killing unit [76]. (**E**) Conjugation of CPPs to antimicrobial peptides enhances membrane permeabilization, membrane translocation, and antibacterial activity [159]. (**F**) Diagrammatic representation of the transport of bifunctional liposomes across the BBB. The glial cells are transfected by the liposomes after they have been moved through the brain’s endothelial cell layer via receptor-mediated transcytosis [162].

## 5. Conclusions

In this brief synopsis, we have concluded that the transmembrane mechanisms of arginine-rich CPPs are concentrated on the application of arginine-rich peptides in biomedicine. In general, the main mechanisms include: (1) guanidine groups of arginine form a bidentate bond with negative phosphates, sulfates, and carboxylates on the cell surface; (2) the charge-neutralized species are driven into the cell by the membrane potential; (3) the guanidine group participates in membrane penetration by partitioning the lipid glycerol regions and attaching to anionic groups on cell surfaces. Significantly, the relationship between the function of arginine-rich peptides and the number of arginine residues, arginine optical isomers, primary sequence, and secondary structure are discussed. Interestingly, arginine-rich CPPs with unique structural features or those linked to hydrophobic hydrocarbon moieties have better membrane penetration rates, which are influenced by modification with different amino acids. A logical design of CPPs will build an effective delivery system, biosensors, and BBB penetration, which exhibits potential to be at clinical level.

## Figures and Tables

**Table 1 ijms-23-09038-t001:** Sequences and nomenclature of the CPPs used in the study.

Name	Sequences	Reference
Tat protein (residues 49–57)	RKKRRQRRR	[26]
R_9_	RRRRRRRRR	[69]
r_9_	rrrrrrrrr	[69]
R_8_	RRRRRRRR	[70]
rR_7_	rRRRRRRR	[70]
(rR)_2_R_4_	rRrRRRRR	[70]
(rR)_3_R_2_	rRrRrRRR	[70]
(rR)_4_	rRrRrRrR	[70]
r_2_(rR)_3_	rrrRrRrR	[70]
r_8_	rrrrrrrr	[70]
W1	LLWRLWRLLWRLRLL	[71]
W5	LLRLLRWWWRLLRLL	[71]
W1-4R	RLLWRLWLWRLLR	[71]
W5-4R	RLLRLLWWWLLRLLR	[71]
CLr	RLLrLLR,	[72]
CL	RLLRLLR	[72]
A2-17	LRKLRKRLLRLWKLRKR	[73]
NP1	stearyl-HHHHHHHHHHHHHHHH-RRRRRRRR-NH_2_	[36,74]
599 peptides	GLFEAIEGFIENGWEGMIDGWYGGGGRRRRRRRRRK	[75]
RR-22	Ac-RGDGPLGLAGI_3_GR_8_-NH_2_	[76]
SynB1	RGGRLSYSRRRFSTSTGR	[77]
R23F	RKKRRQRRRGGSarGVVVHI-Asi-GGKF-NH_2_	[78]
R23DI	RKKRRQRRRGGSarGLTQFGAFIDI-NH_2_	[78]
R23EI	RKKRRQRRRGGSarGVQGLVHISEI-NH_2_	[78]

Note: R = L-arginine; r = D-arginine.

## Data Availability

All relevant data are within the manuscript.

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
