# Peer review of "Membrane Internalization Mechanisms and Design Strategies of Arginine-Rich Cell-Penetrating Peptides"

_ijms, 2022, doi:10.3390/ijms23169038_

Round 1

Reviewer 1 Report

The work represents an interesting review of the use of CPPs at the clinical level. In my opinion, the manuscript is clear and organized. This study summarizes interesting results, and it has the potential to be accepted.

Author Response

Reply: Thank you so much for your positive comments about our work. We further revised our manuscript and corrected the errors/typos and grammar.

Reviewer 2 Report

In the manuscript, the authors describe the properties of arginine-rich cell-penetrating peptides, their cellular entry mechanisms and structure-activity relationships. They also briefly summarize their potential uses. I consider the manuscript a useful overview of this family of cell-penetrating peptides.

 Some remarks:

1. Since a large number of reviews on cell-penetrating peptides have been published, it would be good to present more recent results (not older than 5 years) in sections where it is relevant.

2. Although the sequence of the peptides is given in the text, it would be useful to present them in a summary table (either by merging Table 1 and 2).

3. The ratio between endocytosis and direct cell entry is very important in the cellular uptake of cell-penetrating peptides. It would be good to indicate this for each peptide and that what was the effect of different modifications on it.

4. Reference 52 is incorrectly cited "Tomono et al. revealed that L-octaarginine acquired the penetration-enhancing ability through the conjugation to hyaluronic acid, and the ability disappeared when arginine residues were halved, which once again illustrated that the number of arginine residues was important to the performance of CPPs" (lines 194-197). The tetraarginine derivative showed the same efficacy as the octaarginine derivative.

5. I think these references may increase the value of the review:

Some papers about acylation:

Enhanced Cellular Uptake of Short Polyarginine Peptides through Fatty Acylation and Cyclization, https://doi.org/10.1021/mp500203e

Unsaturated acyl chains dramatically enhanced cellular uptake by direct translocation of a minimalist oligo-arginine lipopeptide, https://doi.org/10.1039/C5CC06116D

Other recent papers about using Dabcyl group as a new kind of modification:

Enhanced Live-Cell Delivery of Synthetic Proteins Assisted by Cell-Penetrating Peptides Fused to DABCYL, doi: 10.1002/anie.202016208.

Influence of the Dabcyl group on the cellular uptake of cationic peptides: short oligoarginines as efficient cell-penetrating peptides, https://doi.org/10.1007/s00726-021-03003-w

Modification of Short Non-Permeable Peptides to Increase Cellular Uptake and Cytostatic Activity of Their Conjugates, https://doi.org/10.1002/slct.202103150

Author Response

Reviewer: 2

Comments:

In the manuscript, the authors describe the properties of arginine-rich cell-penetrating peptides, their cellular entry mechanisms and structure-activity relationships. They also briefly summarize their potential uses. I consider the manuscript a useful overview of this family of cell-penetrating peptides.

Some remarks:

  1. Since a large number of reviews on cell-penetrating peptides have been published, it would be good to present more recent results (not older than 5 years) in sections where it is relevant.

Reply: We gratefully appreciate your valuable suggestion. We have presented more recent references (from 2017) in our paper, and the changes have been marked red in the text.

[3] Haddad, A.F.; Young, J.S.; Aghi, M.K. Using viral vectors to deliver local immunotherapy to glioblastoma. Neurosurgical focus 2021, 50, E4.

[7] Cao, Y.; Ma, E.; Cestellos-Blanco, S.; Zhang, B.; Qiu, R.; Su, Y.; Doudna, J.A.; Yang, P. Nontoxic nanopore electroporation for effective intracellular delivery of biological macromolecules. Proceedings of the National Academy of Sciences 2019, 116, 7899-7904.

[8] Pinheiro, D.H.; Taylor, C.E.; Wu, K.; Siegfried, B.D. Delivery of gene‐specific dsRNA by microinjection and feeding induces RNAi response in Sri Lanka weevil, Myllocerus undecimpustulatus undatus Marshall. Pest management science 2020, 76, 936-943.

[9] Kube, S.; Hersch, N.; Naumovska, E.; Gensch, T.; Hendriks, J.; Franzen, A.; Landvogt, L.; Siebrasse, J.-P.; Kubitscheck, U.; Hoffmann, B. Fusogenic liposomes as nanocarriers for the delivery of intracellular proteins. Langmuir 2017, 33, 1051-1059.

[18] Futaki, S.; Nakase, I. Cell-surface interactions on arginine-rich cell-penetrating peptides allow for multiplex modes of internalization. Accounts of chemical research 2017, 50, 2449-2456.

[19] Kanazawa, T.; Kaneko, M.; Niide, T.; Akiyama, F.; Kakizaki, S.; Ibaraki, H.; Shiraishi, S.; Takashima, Y.; Suzuki, T.; Seta, Y. Enhancement of nose-to-brain delivery of hydrophilic macromolecules with stearate-or polyethylene glycol-modified arginine-rich peptide. International Journal of Pharmaceutics 2017, 530, 195-200.

[22] Kadkhodayan, S.; Bolhassani, A.; Mehdi Sadat, S.; Irani, S.; Fotouhi, F. The efficiency of Tat cell penetrating peptide for intracellular uptake of HIV-1 Nef expressed in E. coli and mammalian cell. Current drug delivery 2017, 14, 536-542.

[34] Böhmová, E.; Machová, D.; Pechar, M.; Pola, R.; Venclíková, K.; Janoušková, O.; Etrych, T. Cell-penetrating peptides: A useful tool for the delivery of various cargoes into cells. Physiological research 2018, 67, S267-S279.

[35] Kardani, K.; Milani, A.; H. Shabani, S.; Bolhassani, A. Cell penetrating peptides: the potent multi-cargo intracellular carriers. Expert opinion on drug delivery 2019, 16, 1227-1258.

[45] Rádis-Baptista, G.; Campelo, I.S.; Morlighem, J.-É.R.; Melo, L.M.; Freitas, V.J. Cell-penetrating peptides (CPPs): From delivery of nucleic acids and antigens to transduction of engineered nucleases for application in transgenesis. Journal of biotechnology 2017, 252, 15-26.

[67] Tai, W.; Gao, X. Functional peptides for siRNA delivery. Advanced drug delivery reviews 2017, 110, 157-168.

  1. Although the sequence of the peptides is given in the text, it would be useful to present them in a summary table (either by merging Table 1 and 2).

Reply: Thank you for your comment. Tables 1 and 2 have been merged into Table 1, and the sequences of the peptides given in the text have been added to Table 1 and marked red.

  1. The ratio between endocytosis and direct cell entry is very important in the cellular uptake of cell-penetrating peptides. It would be good to indicate this for each peptide and that what was the effect of different modifications on it.

Reply: We gratefully appreciate your comment. Yes, we agree that the direct cell entry and endocytosis will be combined to help cargos cross the cell membrane. And the modification of the CPPs will affect the ratio between endocytosis and direct translocation. While the mechanisms of how the modification affects the ratio and the effect of the following details have not been fully understood. The reviewer proposed an exciting point that is worth to investigate further in the future. We have added a discussion section about the critical issue.

Line 64-72: “The two primary recognized cellular uptake mechanisms are endocytosis and the pore formation model. [13,27-35] (Fig. 1A). The ratio between endocytosis and direct cell entry is critical in the cellular uptake of cell-penetrating peptides, and the modification of the CPPs will affect the ratio between endocytosis and direct translocation. While the mechanisms of how the modification affects the ratio and the effect of the following details have not been fully understood. Zhang et al. [36] speculated that the positively charged arginine on the periphery of the NP1 peptides could greatly facilitate their direct translocation through the negatively charged plasma membrane via electrostatic interaction instead of via endocytosis, which provides a more efficient uptake pathway.”

  1. Reference 52 is incorrectly cited "Tomono et al. revealed that L-octaarginine acquired the penetration-enhancing ability through the conjugation to hyaluronic acid, and the ability disappeared when arginine residues were halved, which once again illustrated that the number of arginine residues was important to the performance of CPPs" (lines 194-197). The tetraarginine derivative showed the same efficacy as the octaarginine derivative.

Reply: Thank you so much for your careful check. We have replaced reference 52 with another one, and the new reference was listed as follows. The new example has been added to the text and marked red. 

Line 201-204: “In another study, Naggar et al. revealed that dfR6, dfR7 and dfR8 displayed more robust cytosolic penetration and nucleolar staining compared with dfR4 and dfR5, which once again illustrated that the number of arginine residues was important to the performance of CPPs[63].”

[63]   Najjar, K.; Erazo-Oliveras, A.; Mosior, J.W.; Whitlock, M.J.; Rostane, I.; Cinclair, J.M.; Pellois, J.-P. Unlocking endosomal entrapment with supercharged arginine-rich peptides. Bioconjugate chemistry 2017, 28, 2932-2941.

  1. I think these references may increase the value of the review:

Some papers about acylation:

Enhanced Cellular Uptake of Short Polyarginine Peptides through Fatty Acylation and Cyclization, https://doi.org/10.1021/mp500203e

Unsaturated acyl chains dramatically enhanced cellular uptake by direct translocation of a minimalist oligo-arginine lipopeptide, https://doi.org/10.1039/C5CC06116D

 Other recent papers about using Dabcyl group as a new kind of modification:

Enhanced Live-Cell Delivery of Synthetic Proteins Assisted by Cell-Penetrating Peptides Fused to DABCYL, doi: 10.1002/anie.202016208.

Influence of the Dabcyl group on the cellular uptake of cationic peptides: short oligoarginines as efficient cell-penetrating peptides, https://doi.org/10.1007/s00726-021-03003-w

Modification of Short Non-Permeable Peptides to Increase Cellular Uptake and Cytostatic Activity of Their Conjugates, https://doi.org/10.1002/slct.202103150

Reply: Thanks for your comment. The references have been added to the paper and marked red.

Line 345-357: “Acylation was proved an efficient way to improve the cell-penetrating properties of the arginine-rich peptides. The study of Oh et al.[97] demonstrated that a combination of acylation by long chain fatty acids and cyclization on short arginine-containing peptides can improve their cell-penetrating property, possibly through efficient interaction of rigid positively charged R and hydrophobic dodecanoyl moiety with the corresponding residues in the cell membrane phospholipids. On the other hand, the study of Swiecicki et al. also proved that the unsaturated acyl chain promoted for the short oligo-arginine lipopeptide membrane translocation and endocytosis[98].

In recent research, 4-(dimethylaminoazo) benzene-4-carboxylic acid (DABCYL) was used as a new kind of modification because of its structural simplicity, availability along with hydrophobicity, which could assist cell permeability. It has been demonstrated that utilizing the DABCYL group to modify arginine-rich CPPs effectively increases cellular uptake efficacy[99-101].”

Reviewer 3 Report

I found this mini-review interesting and well-written. References are appropriate. I recommend publication.

Author Response

(The authors gave the same response as above.)

Reviewer 4 Report

The manuscript is well written and addresses important questions about the mechanism of action of arginine-rich CPPs as well as strategies for developing such CPPs.

Small remarks:

1) The authors chose to consider only the features of arginine residues in CPPs, although some of the CPPs they cite use both arginine and lysine. While they go on to explain that arginine is more interesting due to the guanidine group that binds to phosphate, sulfate, and carboxylate groups on the cell surface, they don't explain why some CPPs don't just contain arginine, maybe lysine residues can be dispensed with altogether? It would be correct to mention the functional features of lysine residues in some CPPs and why the authors do not focus on them, maybe they are inferior to arginine residues in the efficiency of such a mechanism of binding to cells compared to arginines?

2) There are citation errors - it would be more correct to use surnames, not first names, as on Line 57 you need Jin et al, not Li et al., the same on Line 93 - Chen, not Xiaochao, Lines 98, 131, 136, 142, 147, 156, 168, 174, 185, 201, 206, 216, 220, 221, 229, 236, 240, 260, 265, 273, 278, 282, 300, 306, 335, 346, 351, 360, 365, 384, 396, 407.425, 428, 433, 464, 499-500, 509.

3) Recently, 3 articles were published in which antimicrobial peptides included CPP from HIV Tat (RKKRRQRRR), and these peptides showed antimicrobial activity comparable to that of antibiotics: PMID: 35008951, PMID: 34575940, PMID: 32887478. This should be mentioned and added to the review.

4) Check if permissions from other authors are needed to use the authors' drawings as part of figures 1, 2, 3 in the manuscript.

Author Response

Reviewer 4:

Comments:

The manuscript is well written and addresses important questions about the mechanism of action of arginine-rich CPPs as well as strategies for developing such CPPs.

  1. The authors chose to consider only the features of arginine residues in CPPs, although some of the CPPs they cite use both arginine and lysine. While they go on to explain that arginine is more interesting due to the guanidine group that binds to phosphate, sulfate, and carboxylate groups on the cell surface, they don't explain why some CPPs don't just contain arginine, maybe lysine residues can be dispensed with altogether? It would be correct to mention the functional features of lysine residues in some CPPs and why the authors do not focus on them, maybe they are inferior to arginine residues in the efficiency of such a mechanism of binding to cells compared to arginines?

Reply: We gratefully appreciate your valuable comments. Wender et al.[1] explained that the presence of multiple guanidine groups in arginine-rich cell-penetrating peptides played a crucial role in membrane penetration ability. Despite having a similar net positive charge number as arginine, lysine had a lower capacity to penetrate membranes because it lacked guanidine groups. Meanwhile, our paper focused on the role of arginine in the cell-penetrating property, so we do not mention the functional features of lysine residues.

[1] Wender PA, Galliher WC, Goun EA, et al. The design of guanidinium-rich transporters and their internalization mechanisms. Adv Drug Deliv Rev, 2008, 60(4/5): 452–472.

  1. There are citation errors - it would be more correct to use surnames, not first names, as on Line 57 you need Jin et al, not Li et al., the same on Line 93 - Chen, not Xiaochao, Lines 98, 131, 136, 142, 147, 156, 168, 174, 185, 201, 206, 216, 220, 221, 229, 236, 240, 260, 265, 273, 278, 282, 300, 306, 335, 346, 351, 360, 365, 384, 396, 407.425, 428, 433, 464, 499-500, 509.

Reply: Thank you so much for your careful check. The errors have been corrected and marked red in the manuscript.

  1. Recently, 3 articles were published in which antimicrobial peptides included CPP from HIV Tat (RKKRRQRRR), and these peptides showed antimicrobial activity comparable to that of antibiotics: PMID: 35008951, PMID: 34575940, PMID: 32887478. This should be mentioned and added to the review.

Reply: Thank you for your suggestions. The part of the antimicrobial peptides that included CPP has been added to the paper (line 501-516) and marked red. Meanwhile, Fig. 3E and Fig. 3F were added to the paper.

Line 513-529: “4.3 Application of Arginine-rich Peptides in Antimicrobial. Due to the quick and broad-spectrum antibacterial activity and apparent decreased risk of microbial resistance development, antimicrobial peptides (AMPs) have drawn much attention as novel families of antibiotics. Interestingly, recent findings suggest that CPP conjugation to AMPs may be useful for increasing antimicrobial activity and selectivity against bacteria. The study of Lee et al.[157] was the first study to investigate the effects of CPP−AMP conjugates on antimicrobial activity as well as their mode of action. They found that CPP (R9) conjugation to AMPs facilitated translocation across the membrane and entry into bacterial cells, and the conjugates showed stronger anti-inflammatory activity than the AMPs alone (Fig. 3E). In the study of Kravchenko et al.[158], hybrid peptides R23F, R23DI, and R23EI (sequences in Table 1) based on the ribosomal S1 protein sequence from S. aureus and with an arginine-rich CPP fragment (RKKRRQRRR) in the N-terminus of the peptides were synthesized and showed antibacterial activity. In addition, Grishin et al.[159] and Kurpe [160] also designed and synthesized hybrid peptides containing arginine-rich CPP fragments in the N-terminus of the peptides, which opened up new possibilities for the manifestation of the antimicrobial effects of hybrid peptides.”

Line 508-512: “(E) Conjugation of CPPs to antimicrobial peptides enhances membrane permeabilization, mem-brane translocation and antibacterial activity [157]. (F) Diagrammatic representation of the transport of bifunctional liposomes across the BBB. The glial cells are transfected by the liposomes after they have been moved through the brain's endothelial cell layer via receptor-mediated transcytosis [162].”

  1. Check if permissions from other authors are needed to use the authors' drawings as part of figures 1, 2, 3 in the manuscript.

Reply: Thank you for your comments. The permissions to use the authors' drawings as part of figures 1, 2, and 3 in the manuscript have been obtained from the publishers.
